# Predicting in-hospital mortality in ICU patients with Coronary heart disease and diabetes mellitus using machine learning models

Guang Tu[1], Zhonglan Cai[1], Ling Wu[2], Hang Yu[3], Hongke Jiang[4], Haijian Luo[5]*

1 Department of Cardiology, Lichuan People's Hospital, Fuzhou, China, 2 Pharmacy Department, The First Affiliated Hospital of Jinan University, Guangzhou, China, 3 Department of Cardiology, The First Affiliated Hospital of Xi 'an Jiaotong University, Xi 'an, China, 4 Department of Physical Education, Shanghai Maritime University, Shanghai, China, 5 Department of Cardiovascular Medicine, Shanghai Public Health Clinical Center (Fudan University), Shanghai, China

* luohaijian@shaphc.org

## Abstract

### Background

Coronary heart disease (CHD) and diabetes mellitus are highly prevalent in intensive care units (ICUs) and significantly contribute to high in-hospital mortality rates. Traditional risk stratification models often fail to capture the complex interactions among clinical variables, limiting their ability to accurately identify high-risk patients. Machine learning (ML) models, with their capacity to analyze large datasets and identify intricate patterns, provide a promising alternative for improving mortality prediction accuracy.

### Objective

This study aims to develop and validate machine learning models for predicting in-hospital mortality in ICU patients with CHD and diabetes, and enhance model interpretability using SHapley Additive exPlanation (SHAP) values, thereby providing a more accurate and practical tool for clinicians.

### Methods

We conducted a retrospective cohort study using data from the MIMIC-IV database, focusing on adult ICU patients with a primary diagnosis of CHD and diabetes. We extracted baseline characteristics, laboratory parameters, and clinical outcomes. The Boruta algorithm was employed for feature selection to identify variables significantly associated with in-hospital mortality, and 16 machine learning models, including logistic regression, random forest, gradient boosting, and neural networks, were developed and compared using receiver operating characteristic (ROC) curves and area under the curve (AUC) analysis. SHAP values were used to explain variable importance and enhance model interpretability.

**Data availability statement:** The data used in this study are available from the Massachusetts Institute of Technology (MIT) and Beth Israel Deaconess Medical Center (BIDMC) upon request. Data can be accessed through the MIMIC-IV v3.1 database, which is publicly available at https://physionet.org/content/mim-iciv/3.1/. Researchers interested in using the data should follow the instructions provided on the website to obtain access.

**Funding:** The author(s) received no specific funding for this work.

**Competing interests:** The authors have declared that no competing interests exist.

**Abbreviations:** CHD: Coronary heart disease; AKI: Acute kidney injury; AUC: Area under the curve; CKD: Chronic kidney disease; GUI: Graphical user interface; ICU: Intensive care unit; INR: International normalized ratio; IQR: Interquartile range; LASSO: Least absolute shrinkage and selection operator; MIMIC-IV: Medical Information Mart for Intensive Care IV; ML: Machine learning; ROC: Receiver operating characteristic; SD: Standard deviation; SHAP: SHapley Additive exPlanations; WBC: White blood cell count.

## Results

Our study included 2,213 patients, of whom 345 (15.6%) experienced in-hospital mortality. The Boruta algorithm identified 29 significant risk factors, and the top 13 variables were used for developing machine learning models. The gradient boosting classifier achieved the highest AUC of 0.8532, outperforming other models. SHAP analysis highlighted age, blood urea nitrogen, and pH as the most important predictors of mortality. SHAP waterfall plots provided detailed individualized risk assessments, demonstrating the model's ability to identify high-risk subgroups effectively.

## Conclusions

Machine learning models, especially the gradient boosting classifier, demonstrated superior performance in predicting in-hospital mortality in ICU patients with CHD and diabetes, outperforming traditional statistical methods. These models provide valuable insights for risk stratification and have the potential to improve clinical outcomes. Future work should focus on external validation and clinical implementation to further enhance their applicability and effectiveness in managing this high-risk population.

## Introduction

Coronary heart disease (CHD) is a leading cause of death and disability globally, with a complex pathogenesis involving plaque rupture, thrombosis, and myocardial ischemia-reperfusion injury [1,2]. The prognosis of CHD remains poor when it is complicated by other diseases, with a significant increase in mortality risk [3,4]. Diabetes mellitus is a common comorbidity in CHD patients, and the interplay between these conditions further deteriorates outcomes [5,6]. In the intensive care unit (ICU) setting, patients with CHD and diabetes are critically ill, with high mortality rates influenced by multiple factors [7]. Early identification of high-risk patients and optimization of treatment strategies are therefore essential.

To address the high mortality risk in these patients, clinicians have relied on risk stratification tools based on clinical experience and statistical methods (e.g., GRACE and TIMI scores) to assess the prognosis of CHD patients [8,9]. However, these tools have limitations in handling complex clinical scenarios and fail to capture the complex interactions between variables. In recent years, machine learning techniques have been widely applied in the medical field to analyze large datasets and identify patterns to predict outcomes more accurately [10–12]. These techniques can handle non-linear relationships and integrate multiple variables to provide more accurate prognostic assessments. However, the application of machine learning in clinical practice still faces challenges related to data quality and model interpretability.

Despite the potential benefits of machine learning, there is a research gap in its application for predicting in-hospital mortality specifically in ICU patients with CHD and diabetes. Previous studies have explored the use of machine learning in predicting mortality in various clinical settings, but few have focused on this high-risk patient

population [13,14]. This study aims to fill this gap by developing and validating machine learning models for predicting in-hospital mortality in ICU patients with CHD and diabetes, enhancing model interpretability using SHapley Additive exPlanation (SHAP) values. By leveraging data from the MIMIC-IV database, we developed and compared multiple machine learning models to provide clinicians with more accurate and practical risk assessment tools to improve patient outcomes.

## Methods

### Data source and study design

This retrospective cohort study utilized data from the MIMIC-IV database, version 3.1, a publicly available, de-identified electronic health record database containing comprehensive clinical data from adult ICU patients admitted to Beth Israel Deaconess Medical Center in Boston, USA [15,16]. Author Guang Tu finished the CITI Data or Specimens Only Research course, obtained approval for database access, and assumed responsibility for data extraction (certification number 65828445). The study included adult ICU patients with a primary diagnosis of CHD and diabetes. CHD was diagnosed based on International Classification of Diseases, Tenth Revision (ICD-10) codes I2510, I25110, I25119, I25118, I25111, and diabetes was diagnosed based on ICD-10 codes E10-E14. Exclusion criteria included: (1) excluded all records with missing values for any of the variables included in the analysis; (2) hospital stay less than 24 hours; (3) age under 18 years. The included patient data covered hospitalizations from 2008 to 2019. (Fig 1)

### Data collection and processing

Baseline characteristics, laboratory parameters, and clinical outcomes were extracted from the MIMIC-IV database, version 3.1. Specific variables included demographic information (e.g., age, sex), physiological parameters (e.g., heart rate, blood pressure), laboratory test results (e.g., blood cell counts, glucose, electrolyte levels, renal function indicators), and comorbidities (e.g., heart failure, diabetes, chronic lung disease). Data preprocessing included handling missing values and variable standardization to ensure the accuracy and stability of model training. All missing values for variables were directly excluded.

### Feature selection

To identify risk factors significantly associated with in-hospital mortality, the Boruta algorithm was employed for feature selection. The Boruta algorithm, a feature selection method based on random forests, automatically identifies features that significantly impact the dependent variable. Regularization parameters ($\lambda$) were determined through cross-validation, and the feature selection process was visualized using coefficient paths and cross-validation plots [17]. Ultimately, the Boruta algorithm identified 29 key featurers, with the top 13 features used for subsequent ML model development. The tentative attributes (marked as yellow in Fig 3) were excluded to ensure the model's simplicity and interpretability.

### Machine learning model development

During the model development phase, we optimized several machine learning models using grid search tuning. Based on the key risk factors identified by the Boruta algorithm, 16 ML models were developed, including logistic regression, random forest, gradient boosting, neural networks, and Extra Trees classifiers. To further evaluate the performance and interpretability of the models, we calculated permutation importance for both the Gradient Boosting Classifier and the Random Forest Classifier using the permutation importance function from scikit-learn. This analysis will help us compare the strengths and weaknesses of these two models in terms of variable importance and predictive power. Each model was trained and validated through cross-validation to ensure its generalizability. Model performance was assessed using receiver operating characteristic (ROC) curves and area under the curve (AUC) analysis, with higher AUC values indicating stronger predictive capabilities. Additionally, accuracy, F1 score, and predictive power were calculated for each model to provide a comprehensive evaluation of performance.

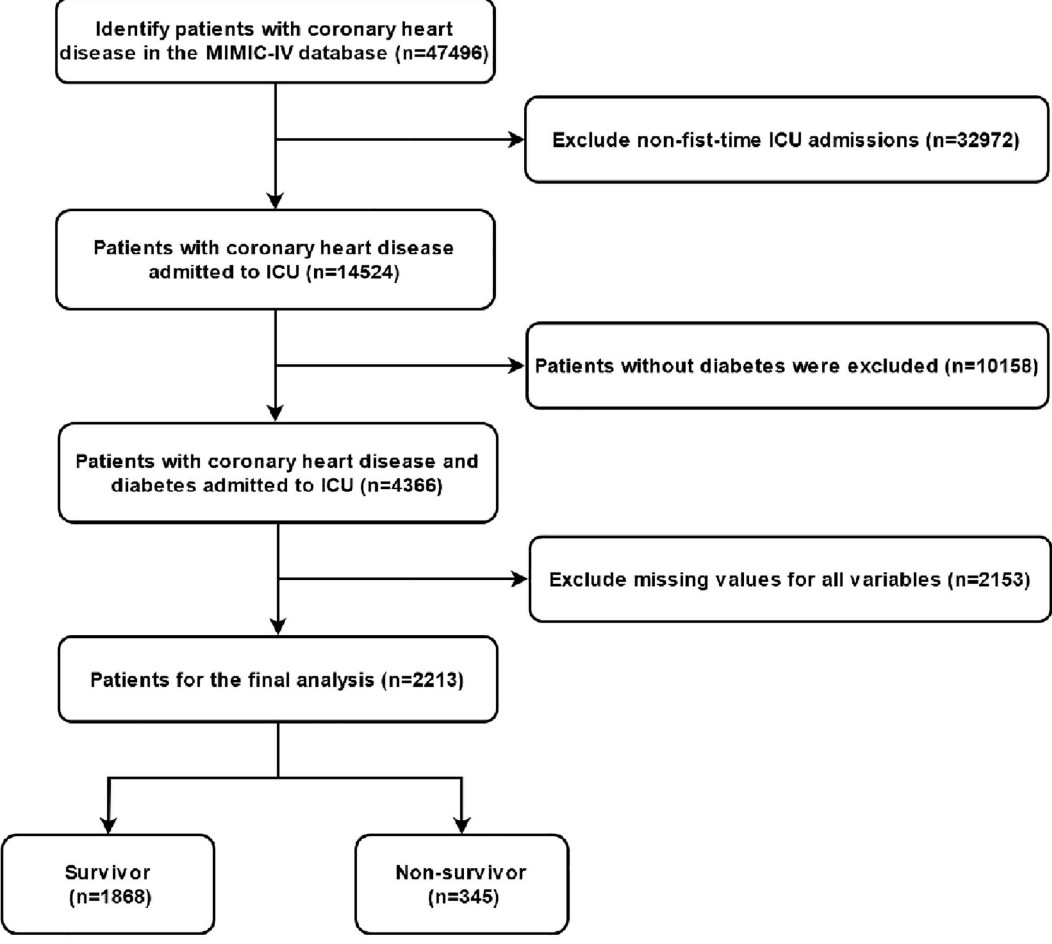

**Fig 1. Flowchart of patient inclusion.**

## Model interpretation and validation

To assess the overall contribution of each variable to the model's predictive performance, we additionally calculated permutation importance using the permutation importance function from scikit-learn (version 1.3.0). This method measures the decrease in model performance (AUC) when the values of a single variable are randomly shuffled, thereby quantifying the strength of association between each predictor and the outcome. To enhance the clinical interpretability of the models, SHAP values were used for model explanation. SHAP values, based on game theory, quantify the contribution of each variable to model predictions and reveal complex interactions between variables. SHAP analysis not only validated the key variables identified by the Boruta algorithm but also assessed their specific impact on individual predictions. Furthermore, SHAP waterfall plots decomposed individual patient predictions, clearly showing the positive or negative contributions of each variable to the prediction results, providing clinicians with tools for individualized risk assessment.

## Workflow of the study

The study's workflow, as depicted in Fig 2, began with data extraction from the MIMIC-IV database, focusing on baseline characteristics, laboratory parameters, and clinical outcomes. This was followed by a preprocessing phase where we

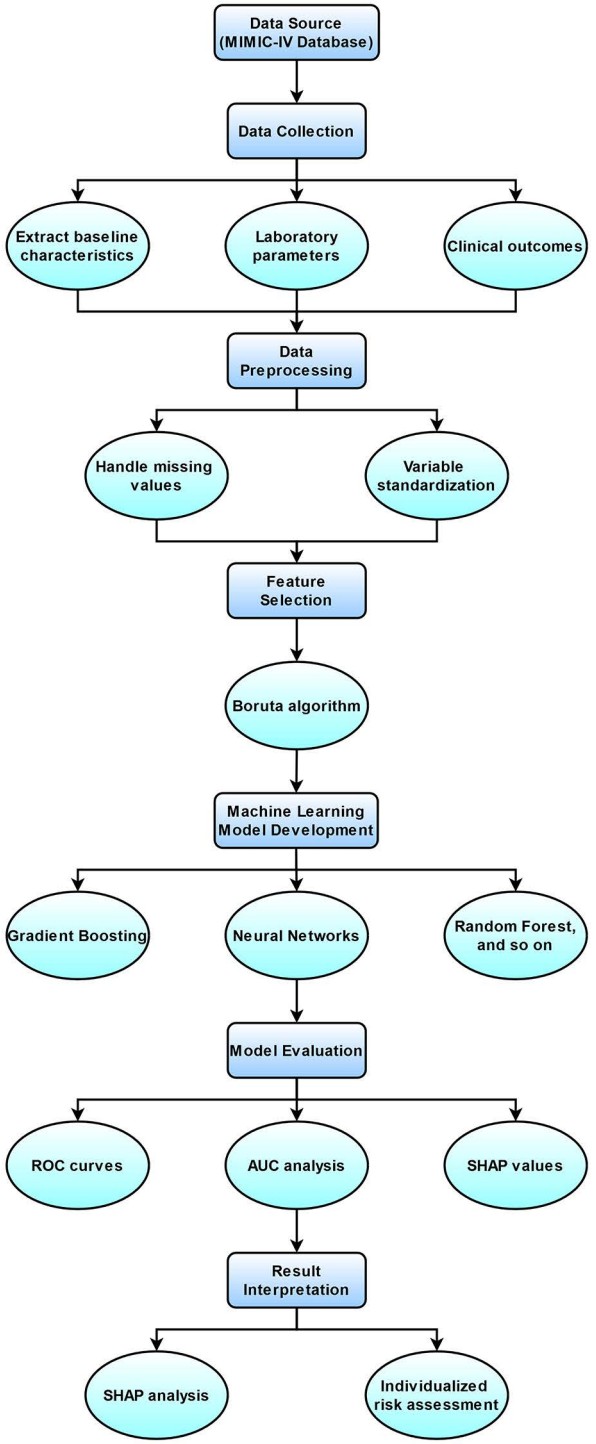

**Fig 2. Workflow diagram.**

handled missing values and performed variable standardization to ensure data quality. Subsequently, feature selection was conducted using the Boruta algorithm to identify key features impacting model predictions. A suite of 16 machine learning models, including Gradient Boosting, Neural Networks, and Random Forest, was then developed. These models were evaluated using ROC curves, AUC analysis, and SHAP values to assess their predictive performance. Finally, we interpreted the results through SHAP analysis and individualized risk assessment to understand the models' predictive mechanisms and the contribution of each feature to the outcomes [18].

## Statistical analysis

Statistical analyses were performed utilizing R Statistical Software (Version 4.2.2; [The R Project for Statistical Computing] (http://www.R-project.org), The R Foundation) and the Free Statistics Analysis Platform (Version 2.1.1; Beijing, China; [http://www.clinicalscientists.cn/freestatistics] (http://www.clinicalscientists.cn/freestatistics)) [19]. Results with a *p*-value below 0.05 were deemed statistically significant. Data processing and model training were conducted on a local server to safeguard data security and privacy.

## Results

### Baseline characteristics

The baseline characteristics of the 2,213 included patients are detailed in Table 1. The table covers demographic information, physiological parameters, laboratory test results, and comorbidities. Significant differences were observed between survivors and non-survivors across multiple variables. The mean age of non-survivors (73.4 years) was significantly higher than that of survivors (68.4 years, $p < 0.001$), indicating that advanced age is an important risk factor for mortality. Additionally, non-survivors had significantly higher systolic and diastolic blood pressures, glucose levels, anion gap, blood urea nitrogen, and creatinine levels, while having lower platelet counts and hemoglobin levels.

### Boruta algorithm selection results

The Boruta algorithm was used to select variables and identify risk factors significantly associated with in-hospital mortality (Fig 3). Based on Boruta analysis, 29 features were selected from the 41 most closely related to in-hospital mortality. The top 13 features, according to Z-values, were used as key input variables for subsequent ML model development, ensuring that the models were built on the most predictive features.

**The horizontal axis is the name of each variable, and the vertical axis is the Z-value of each variable. The box plot shows the Z-value of each variable in the model calculation. The green boxes represent the 29 important variables, the yellow represents tentative attributes, and the red represents unimportant variables.**

### Model prediction performance comparison

In the comparison of model prediction performance, Tables 2 and 3 show that various machine learning models exhibited significant improvements in predicting in-hospital mortality for ICU patients with coronary heart disease and diabetes mellitus after optimization using grid search tuning. Notably, the Gradient Boosting Classifier and the Random Forest Classifier both achieved high performance with AUC values of 0.8532 and 0.8461, respectively. However, their permutation importance and SHAP plots revealed different patterns in variable contributions. Before optimization, the Gradient Boosting Classifier and CatBoost Classifier already demonstrated strong performance, with accuracies of 86.30% and 86.14%, and AUC values of 0.8178 and 0.8150, respectively. After optimization, the Gradient Boosting Classifier achieved the highest performance with an accuracy of 86.51% and an AUC of 0.8528. The CatBoost Classifier followed closely, with an accuracy of 86.57% and an AUC of 0.8509. Other models, including the Ridge Classifier, Logistic Regression, and Random Forest Classifier, also showed notable enhancements in both accuracy and AUC.

**Table 1. Baseline characteristics of the patients.**

| Variables | Total (n = 2213) | Survivors (n = 1868) | Non-survivors (n = 345) | P _value |
|---|---|---|---|---|
| gender, n (%) | | | | 0.002 |
| female | 686 (31.0) | 555 (29.7) | 131 (38) | |
| male | 1527 (69.0) | 1313 (70.3) | 214 (62) | |
| age (year), mean (SD) | 69.2 ± 10.6 | 68.4 ± 10.6 | 73.4 ± 9.9 | < 0.001 |
| heart rate(beats/min), mean (SD) | 69.6 ± 14.2 | 69.5 ± 13.3 | 69.9 ± 18.6 | 0.646 |
| sbp (mmHg), mean (SD) | 86.8 ± 15.6 | 88.1 ± 14.6 | 80.0 ± 19.0 | < 0.001 |
| dbp (mmHg), mean (SD) | 44.4 ± 9.9 | 45.0 ± 9.4 | 41.0 ± 11.8 | < 0.001 |
| spo2, Mean ± SD | 91.2 ± 7.4 | 92.0 ± 5.2 | 87.1 ± 13.4 | < 0.001 |
| hematocrit(mg/dL), Mean ± SD | 28.5 ± 6.2 | 28.4 ± 6.0 | 28.9 ± 7.2 | 0.205 |
| hemoglobin(mg/dL), mean (SD) | 9.2 ± 2.0 | 9.2 ± 2.0 | 9.1 ± 2.3 | 0.558 |
| platelets(×10⁹/L), mean (SD) | 174.5 ± 88.9 | 172.8 ± 86.0 | 183.5 ± 103.0 | 0.040 |
| wbc(×10⁹/L), Median (IQR) | 11.1 ± 6.4 | 10.8 ± 6.0 | 12.9 ± 8.4 | < 0.001 |
| aniongap(mg/dL), mean (SD) | 12.4 ± 4.2 | 11.8 ± 3.8 | 15.6 ± 5.2 | < 0.001 |
| bicarbonate(mmol/dL), Mean ± SD | 20.4 ± 4.9 | 20.9 ± 4.4 | 17.9 ± 6.3 | < 0.001 |
| bun(mg/dL), Median (IQR) | 20.0 (14.0, 34.0) | 18.0 (13.0, 30.0) | 33.0 (21.0, 55.0) | < 0.001 |
| calcium(mmol/dL), mean (SD) | 8.2 ± 0.7 | 8.2 ± 0.7 | 8.1 ± 1.0 | 0.089 |
| chloride(mmol/dL), mean (SD) | 100.4 ± 6.4 | 100.9 ± 6.1 | 98.0 ± 7.2 | < 0.001 |
| creatinine(mg/dL), Median (IQR) | 1.0 (0.8, 1.7) | 1.0 (0.7, 1.4) | 1.7 (1.1, 2.6) | < 0.001 |
| glucose(mmol/dL), mean (SD) | 137.0 ± 56.2 | 134.2 ± 49.8 | 152.1 ± 80.9 | < 0.001 |
| sodium(mmol/dL), mean (SD) | 136.0 ± 4.8 | 136.1 ± 4.6 | 135.5 ± 6.0 | 0.038 |
| potassium(mmol/dL), mean (SD) | 4.1 ± 0.6 | 4.1 ± 0.5 | 4.1 ± 0.7 | 0.319 |
| inr, mean (SD) | 1.4 ± 0.5 | 1.3 ± 0.5 | 1.6 ± 0.7 | < 0.001 |
| Pt (s), mean (SD) | 14.8 ± 5.7 | 14.3 ± 5.0 | 17.6 ± 8.0 | < 0.001 |
| Apt (s), mean (SD) | 31.6 ± 13.4 | 30.5 ± 11.8 | 37.3 ± 18.9 | < 0.001 |
| lactate(mmol/L), Median (IQR) | 1.4 (1.1, 1.9) | 1.4 (1.0, 1.8) | 1.9 (1.3, 3.5) | < 0.001 |
| ph, Mean ± SD | 7.3 ± 0.1 | 7.3 ± 0.1 | 7.3 ± 0.1 | < 0.001 |
| po2, mean (SD) | 93.6 ± 46.1 | 96.1 ± 46.7 | 80.4 ± 40.2 | < 0.001 |
| pco2, mean (SD) | 36.4 ± 8.3 | 36.6 ± 7.9 | 35.7 ± 10.4 | 0.095 |
| myocardial_infarct, n (%) | | | | 0.520 |
| no | 1100 (49.7) | 934 (50) | 166 (48.1) | |
| yes | 1113 (50.3) | 934 (50) | 179 (51.9) | |
| heart_failure, n (%) | | | | < 0.001 |
| no | 1030 (46.5) | 936 (50.1) | 94 (27.2) | |
| yes | 1183 (53.5) | 932 (49.9) | 251 (72.8) | |
| peripheral_vascular, n (%) | | | | < 0.001 |
| no | 1880 (85.0) | 1608 (86.1) | 272 (78.8) | |
| yes | 333 (15.0) | 260 (13.9) | 73 (21.2) | |
| dementia, n (%) | | | | < 0.001 |
| no | 2102 (95.0) | 1789 (95.8) | 313 (90.7) | |
| yes | 111 (5.0) | 79 (4.2) | 32 (9.3) | |
| cerebrovascular, n (%) | | | | < 0.001 |
| no | 1874 (84.7) | 1609 (86.1) | 265 (76.8) | |
| yes | 339 (15.3) | 259 (13.9) | 80 (23.2) | |
| chronic pulmonary disease, n (%) | | | | 0.212 |
| no | 1706 (77.1) | 1449 (77.6) | 257 (74.5) | |
| yes | 507 (22.9) | 419 (22.4) | 88 (25.5) | |

*(Continued)*

**Table 1.** (Continued)

| Variables | Total (n = 2213) | Survivors (n = 1868) | Non-survivors (n = 345) | *P* _value |
|---|---|---|---|---|
| rheumatic disease, n (%) | | | | 0.308 |
| no | 2157 (97.5) | 1818 (97.3) | 339 (98.3) | |
| yes | 56 (2.5) | 50 (2.7) | 6 (1.7) | |
| peptic ulcer disease, n (%) | | | | 0.006 |
| no | 2162 (97.7) | 1832 (98.1) | 330 (95.7) | |
| yes | 51 (2.3) | 36 (1.9) | 15 (4.3) | |
| mild liver disease, n (%) | | | | 0.003 |
| no | 2033 (91.9) | 1730 (92.6) | 303 (87.8) | |
| yes | 180 (8.1) | 138 (7.4) | 42 (12.2) | |
| paraplegia, n (%) | | | | < 0.001 |
| no | 2139 (96.7) | 1821 (97.5) | 318 (92.2) | |
| yes | 74 (3.3) | 47 (2.5) | 27 (7.8) | |
| renal disease, n (%) | | | | < 0.001 |
| no | 1444 (65.3) | 1278 (68.4) | 166 (48.1) | |
| yes | 769 (34.7) | 590 (31.6) | 179 (51.9) | |
| malignant cancer, n (%) | | | | 0.011 |
| no | 2059 (93.0) | 1749 (93.6) | 310 (89.9) | |
| yes | 154 (7.0) | 119 (6.4) | 35 (10.1) | |
| severe liver disease, n (%) | | | | < 0.001 |
| no | 2143 (96.8) | 1820 (97.4) | 323 (93.6) | |
| yes | 70 (3.2) | 48 (2.6) | 22 (6.4) | |
| metastatic solid tumor, n (%) | | | | < 0.001 |
| no | 2157 (97.5) | 1832 (98.1) | 325 (94.2) | |
| yes | 56 (2.5) | 36 (1.9) | 20 (5.8) | |
| aids, n (%) | | | | 1.000 |
| no | 2204 (99.6) | 1860 (99.6) | 344 (99.7) | |
| yes | 9 (0.4) | 8 (0.4) | 1 (0.3) | |

## Machine learning model performance

Fig 4 displays the ROC curves of the 16 ML models. The figure intuitively reflects the models' discriminatory abilities, with higher AUC values indicating better predictive performance. The gradient boosting classifier achieved the highest AUC value (0.8532), demonstrating its best performance in predicting in-hospital mortality. This figure clearly compares the strengths and weaknesses of different models, providing an important basis for selecting the best model.

## Variable importance and SHAP values

Fig 5a displays the permutation importance of the top 10 variables for predicting in-hospital mortality. Fig 5b shows the SHAP summary plot, which evaluates the directionality of these associations. Figs 5a and 5c, and Figs 5b and 5d compare the permutation importance and SHAP summary plots for the Gradient Boosting Classifier and the Random Forest Classifier, respectively. These figures highlight the differences in variable importance and the direction of their effects as identified by each model. Variables such as age, BUN, and pH were assigned higher weights, indicating their key roles in predicting mortality. This analysis helps clinicians understand which factors have the greatest impact on patient prognosis,

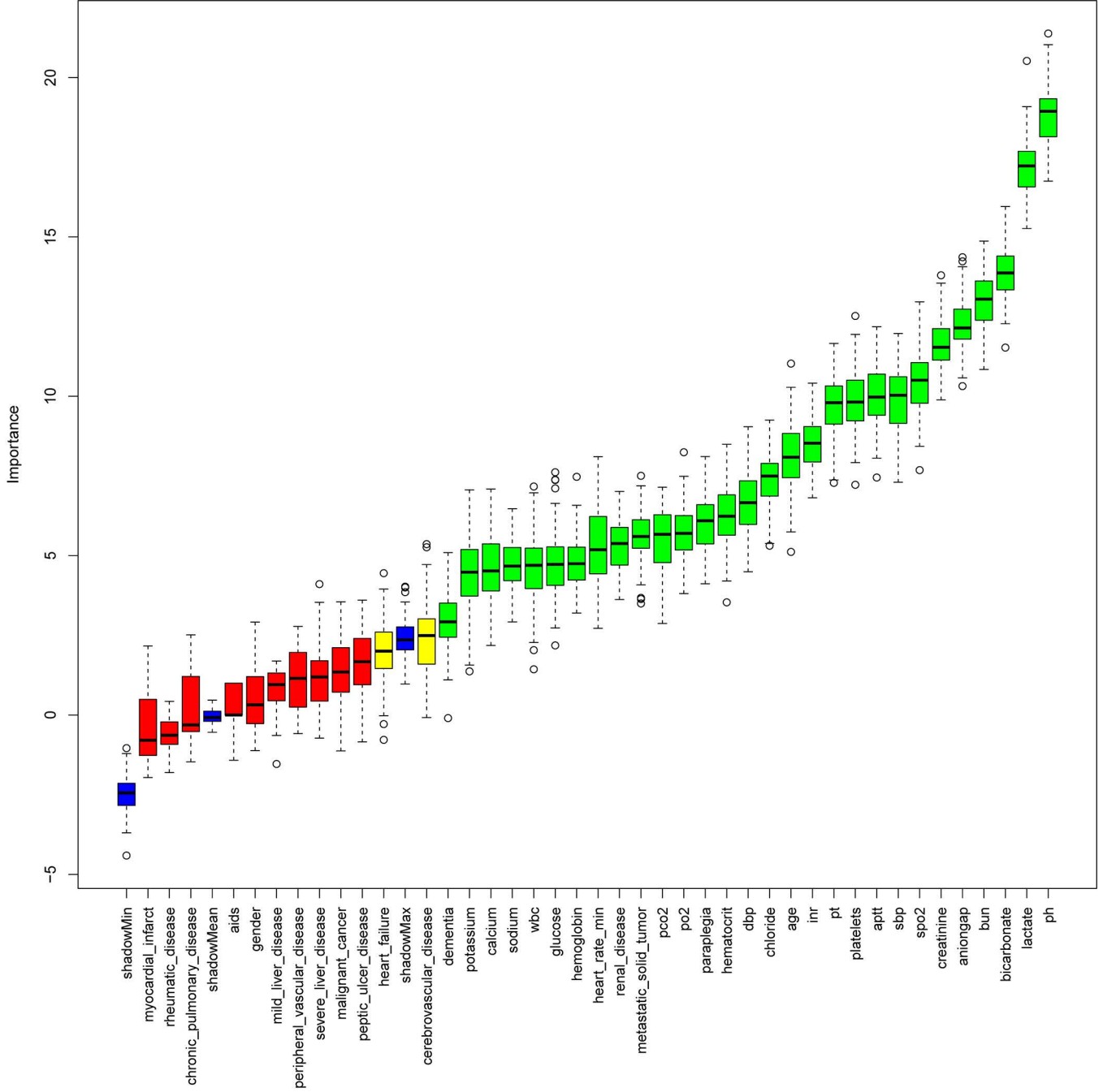

**Fig 3. Feature selection using the Boruta algorithm.**

thereby guiding clinical decision-making. Figs 5b and 5d presents the results of SHAP value analysis. SHAP analysis quantifies the contribution of each variable to model predictions and reveals the specific impact of each feature on individual predictions. This analysis not only validates the key variables identified by the Boruta algorithm (e.g., age, BUN, pH) but also provides details on the interactions between variables.

**Table 2. Comparison of ROC performance before and after optimization.**

| Algorithm | Before optimization | | After optimization | |
|---|---|---|---|---|
| | Accuracy(%) | AUC(%) | Accuracy(%) | AUC(%) |
| Gradient Boosting Classifier | 0.8630 | 0.8178 | 0.8638 | 0.8528 |
| CatBoost Classifier | 0.8614 | 0.8150 | 0.8657 | 0.8509 |
| Ridge Classifier | 0.8569 | 0.5785 | 0.8612 | 0.8466 |
| Logistic Regression | 0.7711 | 0.8115 | 0.8651 | 0.8457 |
| Linear Discriminant Analysis | 0.8554 | 0.8135 | 0.8657 | 0.8451 |
| Light Gradient Boosting Machine | 0.8434 | 0.7909 | 0.8548 | 0.8448 |
| Extra Trees Classifier | 0.8690 | 0.8116 | 0.8528 | 0.8425 |
| Random Forest Classifier | 0.8554 | 0.8148 | 0.8619 | 0.8419 |
| Naive Bayes | 0.8298 | 0.7990 | 0.8186 | 0.8399 |
| Extreme Gradient Boosting | 0.7771 | 0.8059 | 0.8586 | 0.8378 |
| MLP Classifier | 0.8404 | 0.8013 | 0.8554 | 0.8346 |
| Ada Boost Classifier | 0.8373 | 0.7905 | 0.8541 | 0.8278 |
| SVM – Linear Kernel | 0.8057 | 0.7008 | 0.6831 | 0.8117 |
| Gaussian Process Classifier | 0.7907 | 0.6787 | 0.807 | 0.7187 |
| K Neighbors Classifier | 0.8479 | 0.7876 | 0.8483 | 0.7161 |
| Decision Tree Classifier | 0.8524 | 0.7515 | 0.8095 | 0.6349 |

**Table 3. Optimized model prediction performance.**

| Algorithm | AUC(%) | Accuracy(%) | F1score | predictive(%) |
|---|---|---|---|---|
| Gradient Boosting Classifier | 0.8532 | 0.8651 | 0.4322 | 0.6318 |
| CatBoost Classifier | 0.8486 | 0.8644 | 0.4250 | 0.6359 |
| Random Forest Classifier | 0.8461 | 0.8702 | 0.4201 | 0.6996 |
| Light Gradient Boosting Machine | 0.8446 | 0.8573 | 0.4273 | 0.5705 |
| Ridge Classifier | 0.8421 | 0.8618 | 0.3008 | 0.7191 |
| Logistic Regression | 0.8411 | 0.8664 | 0.3920 | 0.6698 |
| Linear Discriminant Analysis | 0.8409 | 0.8664 | 0.4230 | 0.6461 |
| Extreme Gradient Boosting | 0.8399 | 0.8606 | 0.4594 | 0.5796 |
| Naive Bayes | 0.8373 | 0.8238 | 0.4663 | 0.4461 |
| Extra Trees Classifier | 0.8362 | 0.8625 | 0.3540 | 0.6623 |
| MLP Classifier | 0.8344 | 0.8418 | 0.3682 | 0.5465 |
| SVM – Linear Kernel | 0.8293 | 0.8424 | 0.3731 | 0.6038 |
| Ada Boost Classifier | 0.8251 | 0.8490 | 0.4339 | 0.5219 |
| Gaussian Process Classifier | 0.7186 | 0.8070 | 0.3257 | 0.3591 |
| K Neighbors Classifier | 0.7159 | 0.8483 | 0.2848 | 0.5356 |
| Decision Tree Classifier | 0.6307 | 0.8024 | 0.3756 | 0.3707 |

## SHAP waterfall plot

Fig 6 displays the SHAP waterfall plot. The plot decomposes individual patient predictions, clearly showing the positive or negative contributions of each variable to the prediction results. For example, a high systolic blood pressure (SBP) may significantly increase the risk of death, while a lower BUN level may have a protective effect on survival.

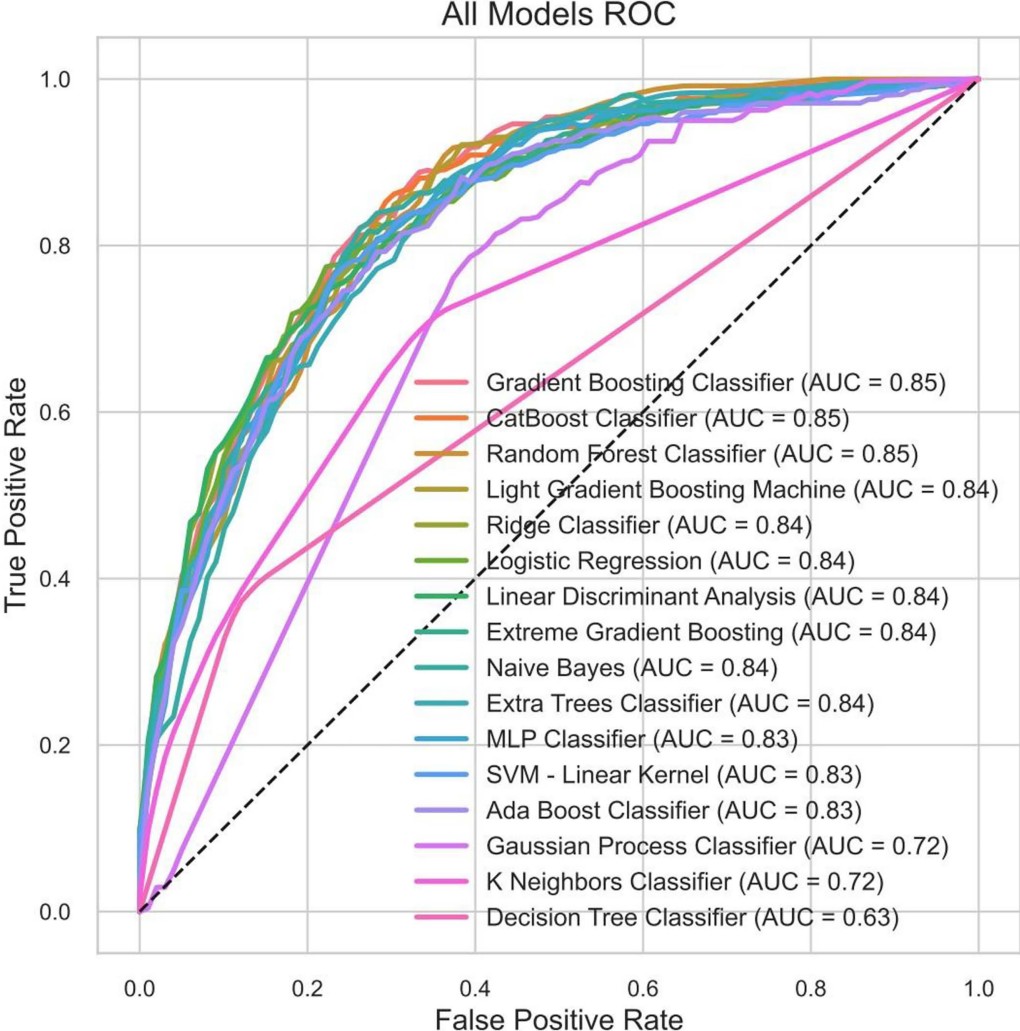

**Fig 4. Receiver operating characteristic curves of 16 models for in-hospital mortality in patients with coronary heart disease with diabetes mellitus.**

## Discussion

This study successfully utilized machine learning models to predict in-hospital mortality in ICU patients with CHD and diabetes. Compared to traditional statistical methods, machine learning models demonstrated a stronger ability to capture complex interactions between clinical variables, thereby significantly improving predictive accuracy [20]. In particular, the gradient boosting classifier achieved the highest AUC value (0.8532), indicating its superior ability to distinguish between high-risk and low-risk patients. This result not only validates the effectiveness of machine learning in handling complex clinical data but also provides clinicians with a powerful tool for early identification of high-risk patients and optimization of treatment strategies, outperforming traditional statistical methods.

The high mortality rate in ICU patients with CHD and diabetes is related to multiple pathophysiological mechanisms. CHD involves coronary plaque rupture, thrombosis, and myocardial ischemia-reperfusion injury, all of which lead to myocardial cell damage and death [21–23]. The presence of diabetes further exacerbates this

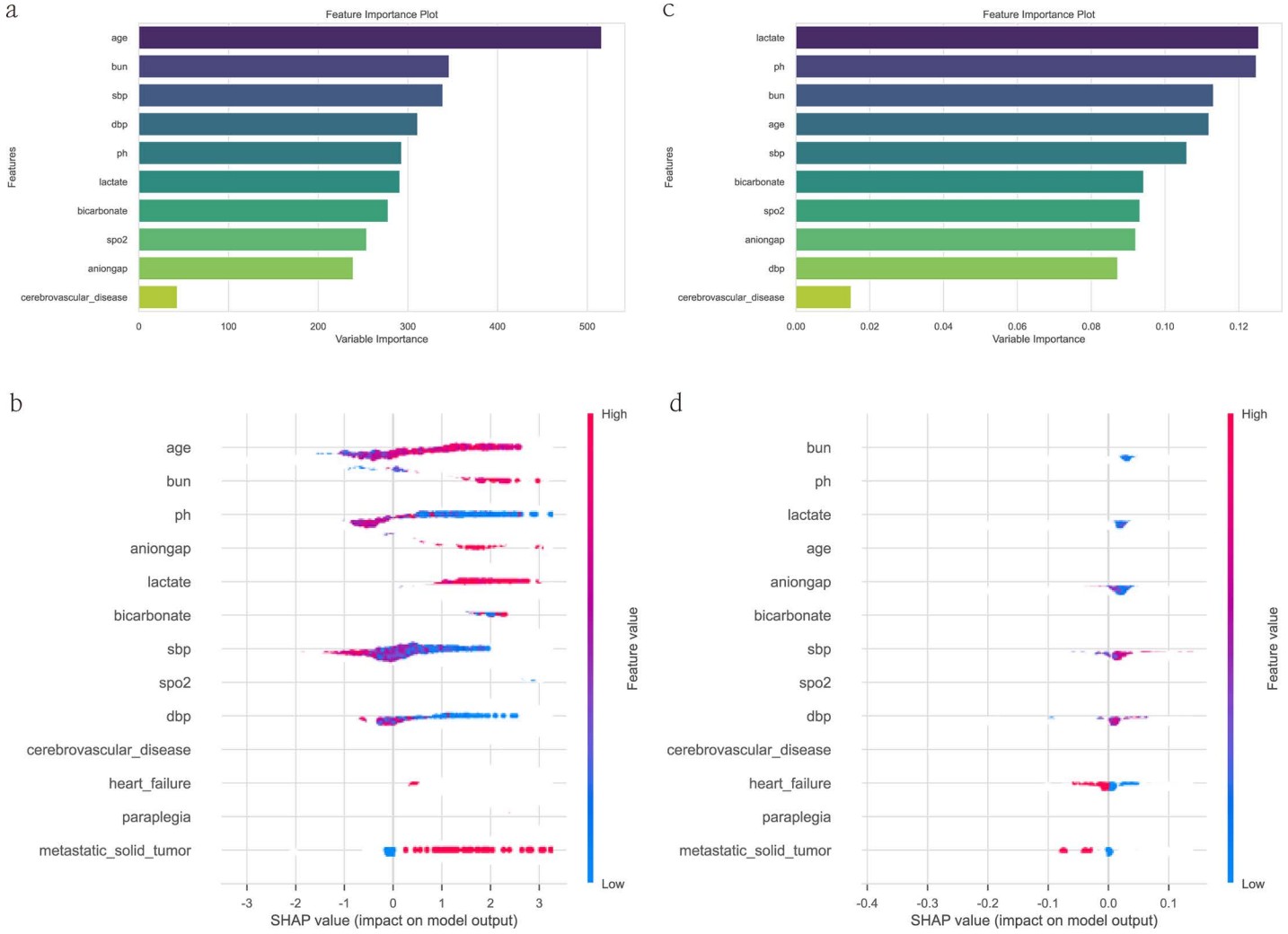

**Fig 5. Permutation importance (a, c) and SHAP summary plots (b, d) for the Gradient Boosting Classifier and Random Forest Classifier, showing variable associations with in-hospital mortality.**

damage. Chronic hyperglycemia in diabetes patients leads to endothelial dysfunction, increased platelet aggregation, and a prothrombotic state, thereby promoting the development of atherosclerosis [24–26]. Additionally, hyperglycemia increases oxidative stress, damaging mitochondrial function in myocardial cells and worsening ischemic injury [27].

From the model results of this study, age, blood urea nitrogen (BUN), and pH were identified as the three most important variables for predicting mortality. The increase in age likely reflects the reduced cardiovascular and metabolic functions in elderly patients, making them less tolerant to the dual insults of CHD and diabetes [28,29]. Elevated BUN levels may indicate renal dysfunction, a common complication in CHD patients, especially those with diabetes. Renal dysfunction leads to the accumulation of toxins, further burdening the heart and affecting myocardial cell metabolism and function [30,31]. Moreover, elevated BUN levels may also be associated with systemic inflammatory responses, which play important roles in the pathogenesis of both CHD and diabetes. Inflammatory factors damage endothelial cells, promote thrombosis, and worsen myocardial ischemic injury [32,33].

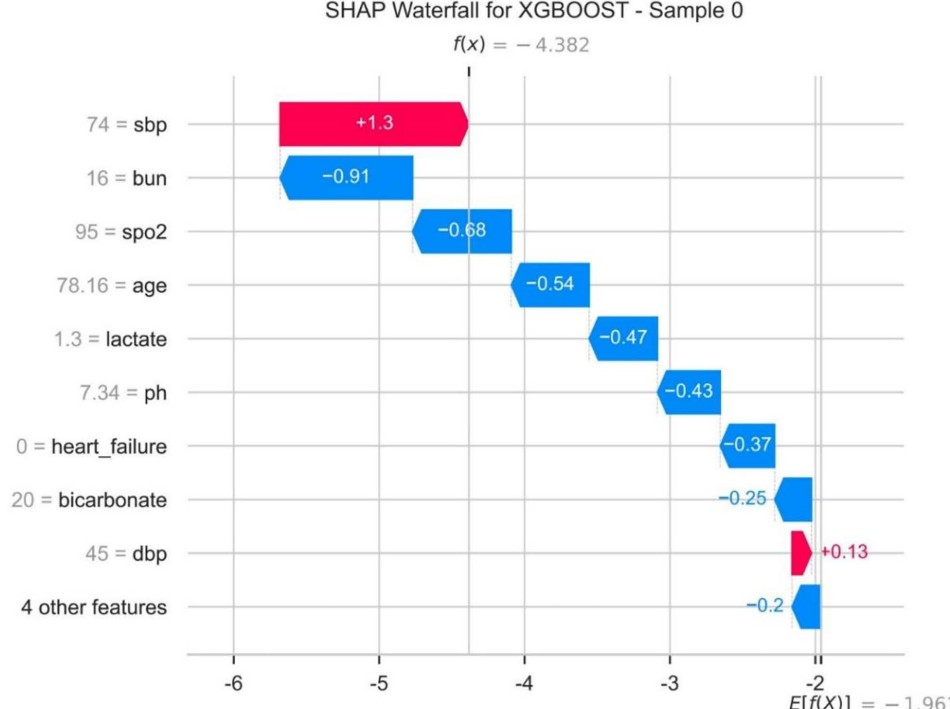

**Fig 6. The SHapley Additive exPlanations (SHAP) waterfall.**

Changes in pH values may be related to the respiratory and circulatory functions of patients. Acidosis (lower pH) may be caused by respiratory failure or metabolic acidosis, which is common in critically ill patients [34,35]. Acidosis affects myocardial contractility, reduces the responsiveness of myocardial cells to catecholamines, and worsens cardiac pump dysfunction. Additionally, acidosis leads to vasodilation, further reducing blood pressure and myocardial perfusion, creating a vicious cycle [36,37].

Other variables such as anion gap, lactate, and bicarbonate were also identified as important risk factors. An elevated anion gap typically indicates the presence of metabolic acidosis, which may be related to lactic acidosis [38]. Elevated lactate levels reflect tissue hypoxia and anaerobic metabolism, important signs of myocardial ischemia and systemic circulatory failure [39]. Changes in bicarbonate levels may be related to the body's compensatory mechanisms for acid-base imbalances, and their abnormalities also suggest severe metabolic disorders in patients [40].

The gradient boosting classifier's superior performance can be attributed to its ability to handle non-linear relationships and integrate a large number of variables. Traditional statistical models often rely on linear assumptions and may fail to fully consider the complex interactions between multiple clinical factors. In comparison, the Random Forest Classifier, while achieving a slightly lower AUC of 0.8461, demonstrated different patterns in permutation importance and SHAP plots. Boosting models, such as Gradient Boosting, focus on the final decision of the strongest tree with minimal bias, making them highly effective in capturing complex interactions. In contrast, Random Forest models rely on the majority vote of all trees with minimal variance, providing robustness against overfitting. The choice between these models depends on the specific characteristics of the dataset and the trade-off between bias and variance. In contrast, machine learning algorithms like gradient boosting can identify subtle patterns in the data that traditional methods cannot detect [17]. For example, in this study, the model identified age, BUN, and pH as key predictive factors, consistent with the pathophysiological mechanisms of CHD and diabetes.

Another significant advantage of machine learning models is their ability to integrate a wide range of clinical variables, including demographic information, physiological parameters, laboratory test results, and comorbidities. This multidimensional data integration capability allows the model to more comprehensively reflect the patient's condition, thereby improving the accuracy and reliability of predictions [10–12]. For example, in this study, in addition to age, BUN, and pH, variables such as anion gap, lactate, and bicarbonate were also identified as important risk factors. The comprehensive consideration of these variables further enhances the model's predictive power, enabling it to more accurately reflect individual patient risks.

### Limitations

However, despite the significant advantages of machine learning models in predictive performance, their application in clinical practice still faces some challenges. First, the complexity of machine learning models may make it difficult for clinicians to understand their decision-making processes. Although SHAP value analysis provides a certain degree of interpretability, how to translate these complex model results into clinically actionable recommendations remains a problem to be solved. Second, the development and validation of machine learning models require large amounts of data and computational resources, which may limit their application in resource-limited healthcare settings. Additionally, the generalizability of the model needs to be further confirmed through external validation to ensure its applicability in different medical environments and patient populations.

### Conclusions

This study successfully utilized machine learning models to predict in-hospital mortality in ICU patients with CHD and diabetes, enhancing model interpretability through SHAP value analysis. These results demonstrate that machine learning models not only outperform traditional statistical methods in predictive accuracy but also provide clinicians with powerful tools for early identification of high-risk patients and optimization of treatment strategies. As technology continues to advance and clinical applications expand, machine learning is expected to play an increasingly important role in future medical practice, providing strong support for improving patient outcomes.

### Supporting information

**S1. Data.**
(XLS)

**S2. MIMIC database.**
(PDF)

### Author contributions

**Conceptualization:** guang tu, Haijian Luo.

**Data curation:** guang tu, Zhonglan Cai, Ling Wu, Hang Yu, Hongke Jiang.

**Formal analysis:** Zhonglan Cai, Ling Wu, Hang Yu, Hongke Jiang.

**Funding acquisition:** Zhonglan Cai.

**Writing – original draft:** guang tu, Haijian Luo.

**Writing – review & editing:** guang tu, Haijian Luo.

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
