## [Decision Letter · Decision Letter 0]

14 May 2025

Dear Dr. tu,

We look forward to receiving your revised manuscript.

Kind regards,

Habtamu Setegn Ngusie

Academic Editor

PLOS ONE

Journal Requirements:

2. Please ensure that you refer to Figure 1 and 2 in your text as, if accepted, production will need this reference to link the reader to the figure.

**Additional Editor Comments:**

As the editor, I would like to offer the following general comments:

Introduction: Please ensure that your introduction is scientifically sound. Start with a general overview and then narrow down to specific details. Incorporate existing solutions to the problem, highlight the research gap, and reference previous studies.

Language Editing: I recommend enhancing the overall quality of the English language in your manuscript. Please check all grammatical issues from the introduction to the end, as this is vital for your and the journal's reputation.

Abstract: Please revise your abstract to make it more compelling and ensure it meets the journal's standards. It should serve as a concise summary of the key sections of your manuscript.

Specific Comments:

Please provide a clear workflow diagram.

Please employ a hyperparameter tuning technique in your study and show the ROC curve before and after tuning/optimization. You can use only one technique, for example, grid search tuning or Bayesian optimization.

You state that you used the Boruta algorithm for risk factor identification; I believe you meant feature selection, as this algorithm is used for that purpose. Please correct your statement in all sections that mention the Boruta algorithm accordingly.

If you obtain new results regarding the best-performing algorithm after employing the hyperparameter tuning technique, please rewrite the results, discussion, and other sections to reflect these new findings.

In your conclusion, you state your findings. Refer to the following quoted sentence from your abstract, conclusion subsection: "These models outperform traditional statistical methods and offer valuable insights for risk stratification." Do you mean compared to other machine learning algorithms or compared to traditional statistical methods? Please clarify and ensure your conclusion aligns with your findings.

In your Boruta algorithm graph (Figure 2), the yellow represents tentative attributes, but you seem to have removed those variables as you did for non-important variables. Should tentative variables be removed automatically like non-important variables? If not, please clearly state what you did in your feature selection technique.

It would be better to incorporate a workflow diagram for your machine learning activities, or if you don't prefer this kind of diagram, please state the details within the methods section.

The performance of each machine learning algorithm, such as the ROC curve value before and after various data balancing techniques, should be highlighted. Please experiment with various data balancing techniques, or if you only employed one, provide the reasoning with references.

Please also highlight headings and subheadings clearly, ensuring the first letter of each heading and subheading is capitalized.

Did you employ SHAP analysis after selecting the most important variables using the top-performing machine learning algorithm, or separately? If it was separately, what was the purpose of employing the machine learning algorithm? If you used both the machine learning algorithm and SHAP analysis for identifying the top predictors, please state the details regarding the method of ensembling.

Lastly, the author may benefit from citing the following article for some of their methodological arguments, as it clearly articulates the aspects we should follow in machine learning, especially in predictive modeling: https://link.springer.com/article/10.1186/s12889-024-19566-8.

Reviewers' comments:

Reviewer's Responses to Questions

**Comments to the Author**

1. Is the manuscript technically sound, and do the data support the conclusions?

Reviewer #1: Partly

2. Has the statistical analysis been performed appropriately and rigorously?

Reviewer #1: I Don't Know

3. Have the authors made all data underlying the findings in their manuscript fully available?

Reviewer #1: No

4. Is the manuscript presented in an intelligible fashion and written in standard English?

Reviewer #1: Yes

Reviewer #1: 1.Unclear inclusion and exclusion criteria can impact sample size and outcomes. Therefore, it is crucial to clarify key factors such as MIMIC database version, age, and diabetes classification to ensure accurate sample size and consistent research baseline.

2.The exclusion criteria only mention CHD patients admitted to the ICU. How should data from CHD patients with multiple ICU admissions be handled? Additionally, how does a change in sample size affect the selection of key variables in the machine learning model?

3.The MIMIC database includes clinical data such as vital signs, laboratory results, clinical scores, complications, medication usage, interventions, and outcomes. The exclusion criteria mentioned in the text only state "lack of key clinical data (e.g., vital signs or laboratory results at admission)" without providing original data to verify potential bias in the exclusion criteria. This lack of transparency may affect the sample size included.

4.Can you provide sufficient evidence that the application of 16 predictive models in clinical prediction research is meaningful, both statistically and clinically?

5.The study mentions the predictive significance of comparing 16 machine learning models. It also examines whether the predictions are consistent with traditional clinical scores such as SOFA, APACHE-II, and OASIS.

6.In our study, age, blood urea nitrogen (BUN), and pH were identified as the three most important prognostic variables. How does this differ from the machine learning model predicting coronary heart disease (CHD) in type 2 diabetes published in Acta Diabetologica on 2025 Apr 1?

Ji Y, Shang H, Yi J, Zang W, Cao W. Machine learning-based models to predict type 2 diabetes combined with coronary heart disease and feature analysis-based on interpretable SHAP. Acta Diabetol. 2025 Apr 1. doi: 10.1007/s00592-025-02496-1. Epub ahead of print. PMID: 40167635.

7.Recently, many studies on CHD outcome prediction models can be found on PUBMED. What do you consider to be the advantages or innovations of this study?

For example:

1、Yadegar A, Mohammadi F, Seifouri K, Mokhtarpour K, Yadegar S, Bahrami Hazaveh E, Seyedi SA, Rabizadeh S, Esteghamati A, Nakhjavani M. Surrogate markers of insulin resistance and coronary artery disease in type 2 diabetes: U-shaped TyG association and insights from machine learning integration. Lipids Health Dis. 2025 Mar 15;24(1):96. doi: 10.1186/s12944-025-02526-5. PMID: 40089748; PMCID: PMC11910848.

2、Rehman MU, Naseem S, Butt AUR, Mahmood T, Khan AR, Khan I, Khan J, Jung Y. Predicting coronary heart disease with advanced machine learning classifiers for improved cardiovascular risk assessment. Sci Rep. 2025 Apr 17;15(1):13361. doi: 10.1038/s41598-025-96437-1. PMID: 40247042; PMCID: PMC12006408.

3、Soleimani H, Najdaghi S, Davani DN, Dastjerdi P, Samimisedeh P, Shayesteh H, Sattartabar B, Masoudkabir F, Ashraf H, Mehrani M, Jenab Y, Hosseini K. Predicting In-Hospital Mortality in Patients With Acute Myocardial Infarction: A Comparison of Machine Learning Approaches. Clin Cardiol. 2025 Apr;48(4):e70124. doi: 10.1002/clc.70124. PMID: 40143742; PMCID: PMC11947610.

4.Tao H, Wang C, Qi H, Li H, Li Y, Xie R, Dai Y, Sun Q, Zhang Y, Yu X, Shen T. A Real-Time Computer-Aided Diagnosis System for Coronary Heart Disease Prediction Using Clinical Information. Rev Cardiovasc Med. 2025 Mar 17;26(3):26204. doi: 10.31083/RCM26204. PMID: 40160568; PMCID: PMC11951285.

**Do you want your identity to be public for this peer review?** For information about this choice, including consent withdrawal, please see our Privacy Policy

Reviewer #1: No

---

## [Author Response · Author response to Decision Letter 1]

23 May 2025

Response to Reviewers

Journal Requirements:

Response:

Thank you very much for your careful review of our manuscript and the valuable comments provided. We have made comprehensive revisions to our manuscript, with particular attention to the format requirements for the title and references. Below are the specific revisions and explanations:

1. Title Section

Before Revision: Predicting In-Hospital Mortality in ICU Patients with Coronary Heart Disease and Diabetes Mellitus Using Machine Learning Models

After Revision: Predicting in-hospital mortality in ICU patients with coronary heart disease and diabetes mellitus using machine learning models

2. References Section

Before Revision:

[1] Fernandez R C, Konate K, Josse E, et al. Therapeutic Peptides to Treat Myocardial Ischemia-Reperfusion Injury[J]. Front Cardiovasc Med, 2022,9:792885.

[2] Welch T D, Yang E H, Reeder G S, et al. Modern management of acute myocardial infarction[J]. Curr Probl Cardiol, 2012,37(7):237-310.

[3] Mureddu G F, D'Errigo P, Rosato S, et al. The relative impact of components of high residual risk on the long-term prognosis after AMI[J]. Int J Cardiol Cardiovasc Risk Prev, 2024,22:200310.

[4] Honda T, Kanazawa H, Koga H, et al. Heart rate on admission is an independent risk factor for poor cardiac function and in-hospital death after acute myocardial infarction[J]. J Cardiol, 2010,56(2):197-203.

[5] Kufazvinei T, Chai J, Boden K A, et al. Emerging opportunities to target inflammation: myocardial infarction and type 2 diabetes[J]. Cardiovasc Res, 2024,120(11):1241-1252.

[6] Xia J G, Li B, Zhang H, et al. Precise Metabolomics Defines Systemic Metabolic Dysregulation Distinct to Acute Myocardial Infarction Associated With Diabetes[J]. Arterioscler Thromb Vasc Biol, 2023,43(4):581-596.

[7] Wernly B, Lichtenauer M, Franz M, et al. Differential Impact of Hyperglycemia in Critically Ill Patients: Significance in Acute Myocardial Infarction but Not in Sepsis?[J]. Int J Mol Sci, 2016,17(9).

[8] Mitarai T, Tanabe Y, Akashi Y J, et al. A novel risk stratification system "Angiographic GRACE Score" for predicting in-hospital mortality of patients with acute myocardial infarction: Data from the K-ACTIVE Registry[J]. J Cardiol, 2021,77(2):179-185.

[9] Roy S S, Abu A S, Khalequzzaman M, et al. GRACE and TIMI risk scores in predicting the angiographic severity of non-ST elevation acute coronary syndrome[J]. Indian Heart J, 2018,70 Suppl 3(Suppl 3):S250-S253.

[10] Zhan K, Buhler K A, Chen I Y, et al. Systemic lupus in the era of machine learning medicine[J]. Lupus Sci Med, 2024,11(1).

[11] Daidone M, Ferrantelli S, Tuttolomondo A. Machine learning applications in stroke medicine: advancements, challenges, and future prospectives[J]. Neural Regen Res, 2024,19(4):769-773.

[12] Galal A, Talal M, Moustafa A. Applications of machine learning in metabolomics: Disease modeling and classification[J]. Front Genet, 2022,13:1017340.

[13] Johnson A E, Stone D J, Celi L A, et al. The MIMIC Code Repository: enabling reproducibility in critical care research[J]. J Am Med Inform Assoc, 2018,25(1):32-39.

[14] Zhou Y, Chen Y, Liang S, et al. Association between potassium fluctuation and in-hospital mortality in acute myocardial infarction patients: a retrospective analysis of the MIMIC-IV database[J]. Clin Res Cardiol, 2025.

[15] Zhou H, Xin Y, Li S. A diabetes prediction model based on Boruta feature selection and ensemble learning[J]. BMC Bioinformatics, 2023,24(1):224.

[16] Han K Y, Gu J, Wang Z, et al. Association Between METS-IR and Prehypertension or Hypertension Among Normoglycemia Subjects in Japan: A Retrospective Study[J]. Front Endocrinol (Lausanne), 2022,13:851338.

[17] Praveen S P, Hasan M K, Abdullah S, et al. Enhanced feature selection and ensemble learning for cardiovascular disease prediction: hybrid GOL2-2 T and adaptive boosted decision fusion with babysitting refinement[J]. Front Med (Lausanne), 2024,11:1407376.

[18] Wang K, Zhu Q, Liu W, et al. Mitochondrial apoptosis in response to cardiac ischemia-reperfusion injury[J]. J Transl Med, 2025,23(1):125.

[19] Chen J, Wang B, Meng T, et al. Oxidative Stress and Inflammation in Myocardial Ischemia-Reperfusion Injury: Protective Effects of Plant-Derived Natural Active Compounds[J]. J Appl Toxicol, 2024.

[20] Młynarska E, Czarnik W, Fularski P, et al. From Atherosclerotic Plaque to Myocardial Infarction-The Leading Cause of Coronary Artery Occlusion[J]. Int J Mol Sci, 2024,25(13).

[21] Yang T, Zhang D. Research progress on the effects of novel hypoglycemic drugs in diabetes combined with myocardial ischemia/reperfusion injury[J]. Ageing Res Rev, 2023,86:101884.

[22] Li Z, Zhang J, Ma Z, et al. Endothelial YAP Mediates Hyperglycemia-Induced Platelet Hyperactivity and Arterial Thrombosis[J]. Arterioscler Thromb Vasc Biol, 2024,44(1):254-270.

[23] Chiva-Blanch G, Peña E, Cubedo J, et al. Molecular mapping of platelet hyperreactivity in diabetes: the stress proteins complex HSPA8/Hsp90/CSK2α and platelet aggregation in diabetic and normal platelets[J]. Transl Res, 2021,235:1-14.

[24] Rudokas M W, McKay M, Toksoy Z, et al. Mitochondrial network remodeling of the diabetic heart: implications to ischemia related cardiac dysfunction[J]. Cardiovasc Diabetol, 2024,23(1):261.

[25] Hajduk A M, Dodson J A, Murphy T E, et al. Risk Model for Decline in Activities of Daily Living Among Older Adults Hospitalized With Acute Myocardial Infarction: The SILVER-AMI Study[J]. J Am Heart Assoc, 2020,9(19):e015555.

[26] Dianati-Maleki N, Butler J. Diabetes Mellitus in Patients With Heart Failure: Bad for All, Worse for Some[J]. JACC Heart Fail, 2017,5(1):25-27.

[27] Kim C S, Choi J S, Bae E H, et al. Association of metabolic syndrome and renal insufficiency with clinical outcome in acute myocardial infarction[J]. Metabolism, 2013,62(5):669-676.

[28] Sriperumbuduri S, Clark E, Hiremath S. New Insights Into Mechanisms of Acute Kidney Injury in Heart Disease[J]. Can J Cardiol, 2019,35(9):1158-1169.

[29] Kozakova M, Morizzo C, Goncalves I, et al. Cardiovascular organ damage in type 2 diabetes mellitus: the role of lipids and inflammation[J]. Cardiovasc Diabetol, 2019,18(1):61.

[30] Kanter J E, Bornfeldt K E. Inflammation and diabetes-accelerated atherosclerosis: myeloid cell mediators[J]. Trends Endocrinol Metab, 2013,24(3):137-144.

[31] Achanti A, Szerlip H M. Acid-Base Disorders in the Critically Ill Patient[J]. Clin J Am Soc Nephrol, 2023,18(1):102-112.

[32] Masevicius F D, Rubatto B P, Risso V A, et al. Relationship of at Admission Lactate, Unmeasured Anions, and Chloride to the Outcome of Critically Ill Patients[J]. Crit Care Med, 2017,45(12):e1233-e1239.

[33] Kimmoun A, Ducrocq N, Levy B. New conclusive data on human myocardial dysfunction induced by acidosis[J]. Crit Care, 2012,16(5):160.

[34] Kumbhani D J, Healey N A, Thatte H S, et al. Intraoperative myocardial acidosis as a risk for hospital readmission after cardiac surgery[J]. Am J Surg, 2009,198(3):373-380.

[35] Romero J E, Htyte N. An unusual cause of high anion gap metabolic acidosis: pyroglutamic acidemia. A case report[J]. Am J Ther, 2013,20(5):581-584.

[36] Zymliński R, Biegus J, Sokolski M, et al. Increased blood lactate is prevalent and identifies poor prognosis in patients with acute heart failure without overt peripheral hypoperfusion[J]. Eur J Heart Fail, 2018,20(6):1011-1018.

[37] Sheikh I A, Malik A, AlBasri S, et al. In silico identification of genes involved in chronic metabolic acidosis[J]. Life Sci, 2018,192:246-252.

After Revision:

References

[1] Fernandez RC, Konate K, Josse E, Nargeot J, Barrère-Lemaire S, Boisguérin P. Therapeutic Peptides to Treat Myocardial Ischemia-Reperfusion Injury[J]. Front Cardiovasc Med, 2022,9:792885.

[2] Welch TD, Yang EH, Reeder GS, Gersh BJ. Modern management of acute myocardial infarction[J]. Curr Probl Cardiol, 2012,37(7):237-310.

[3] Mureddu GF, D'Errigo P, Rosato S, Faggiano P, Badoni G, Ceravolo R, et al. The relative impact of components of high residual risk on the long-term prognosis after AMI[J]. Int J Cardiol Cardiovasc Risk Prev, 2024,22:200310.

[4] Honda T, Kanazawa H, Koga H, Miyao Y, Fujimoto K. Heart rate on admission is an independent risk factor for poor cardiac function and in-hospital death after acute myocardial infarction[J]. J Cardiol, 2010,56(2):197-203.

[5] Kufazvinei T, Chai J, Boden KA, Channon KM, Choudhury RP. Emerging opportunities to target inflammation: myocardial infarction and type 2 diabetes[J]. Cardiovasc Res, 2024,120(11):1241-1252.

[6] Xia JG, Li B, Zhang H, Li QX, Lam SM, Yin CL, et al. Precise Metabolomics Defines Systemic Metabolic Dysregulation Distinct to Acute Myocardial Infarction Associated With Diabetes[J]. Arterioscler Thromb Vasc Biol, 2023,43(4):581-596.

[7] Wernly B, Lichtenauer M, Franz M, Kabisch B, Muessig J, Masyuk M, et al. Differential Impact of Hyperglycemia in Critically Ill Patients: Significance in Acute Myocardial Infarction but Not in Sepsis?[J]. Int J Mol Sci, 2016,17(9).

[8] Mitarai T, Tanabe Y, Akashi YJ, Maeda A, Ako J, Ikari Y, et al. A novel risk stratification system "Angiographic GRACE Score" for predicting in-hospital mortality of patients with acute myocardial infarction: Data from the K-ACTIVE Registry[J]. J Cardiol, 2021,77(2):179-185.

[9] Roy SS, Abu AS, Khalequzzaman M, Ullah M, Arifur RM. GRACE and TIMI risk scores in predicting the angiographic severity of non-ST elevation acute coronary syndrome[J]. Indian Heart J, 2018,70 Suppl 3(Suppl 3):S250-S253.

[10] Zhan K, Buhler KA, Chen IY, Fritzler MJ, Choi MY. Systemic lupus in the era of machine learning medicine[J]. Lupus Sci Med, 2024,11(1).

[11] Daidone M, Ferrantelli S, Tuttolomondo A. Machine learning applications in stroke medicine: advancements, challenges, and future prospectives[J]. Neural Regen Res, 2024,19(4):769-773.

[12] Galal A, Talal M, Moustafa A. Applications of machine learning in metabolomics: Disease modeling and classification[J]. Front Genet, 2022,13:1017340.

[13] Pieszko K, Hiczkiewicz J, Budzianowski P, Budzianowski J, Rzeźniczak J, Pieszko K, et al. Predicting Long-Term Mortality after Acute Coronary Syndrome Using Machine Learning Techniques and Hematological Markers[J]. Dis Markers, 2019,2019:9056402.

[14] Ermak AD, Gavrilov DV, Novitskiy RE, Gusev AV, Andreychenko AE. Development, evaluation and validation of machine learning models to predict hospitalizations of patients with coronary artery disease within the next 12 months[J]. Int J Med Inform, 2024,188:105476.

[15] Johnson AE, Stone DJ, Celi LA, Pollard TJ. The MIMIC Code Repository: enabling reproducibility in critical care research[J]. J Am Med Inform Assoc, 2018,25(1):32-39.

[16] Zhou Y, Chen Y, Liang S, Li Y, Zhao C, Wu Z. Association between potassium fluctuation and in-hospital mortality in acute myocardial infarction patients: a retrospective analysis of the MIMIC-IV database[J]. Clin Res Cardiol, 2025.

[17] Zhou H, Xin Y, Li S. A diabetes prediction model based on Boruta feature selection and ensemble learning[J]. BMC Bioinformatics, 2023,24(1):224.

[18] Ngusie HS, Mengiste SA, Zemariam AB, Molla B, Tesfa GA, Seboka B, et al. Predicting adverse birth outcome among childbearing women in Sub-Saharan Africa: employing innovative machine learning techniques[J]. BMC Public Health, 2024,24(1):2029.

[19] Han KY, Gu J, Wang Z, Liu J, Zou S, Yang CX, et al. Association Between METS-IR and Prehypertension or Hypertension Among Normoglycemia Subjects in Japan: A Retrospective Study[J]. Front Endocrinol (Lausanne), 2022,13:851338.

[20] Praveen SP, Hasan MK, Abdullah S, Sirisha U, Tirumanadham N, Islam S, et al. Enhanced feature selection and ensemble learning for cardiovascular disease prediction: hybrid GOL2-2 T and adaptive boosted decision fusion with babysitting refinement[J]. Front Med (Lausanne), 2024,11:1407376.

[21] Wang K, Zhu Q, Liu W, Wang L, Li X, Zhao C, et al. Mitochondrial apoptosis in response to cardiac ischemia-reperfusion injury[J]. J Transl Med, 2025,23(1):125.

[22] Chen J, Wang B, Meng T, Li C, Liu C, Liu Q, et al. Oxidative Stress and Inflammation in Myocardial Ischemia-Reperfusion Injury: Protective Effects of Plant-Derived Natural Active Compounds[J]. J Appl Toxicol, 2024.

[23] Młynarska E, Czarnik W, Fularski P, Hajdys J, Majchrowicz G, Stabrawa M, et al. From Atherosclerotic Plaque to Myocardial Infarction-The Leading Cause of Coronary Artery Occlusion[J]. Int J Mol Sci, 2024,25(13).

[24] Yang T, Zhang D. Research progress on the effects of novel hypoglycemic drugs in diabetes combined with myocardial ischemia/reperfusion injury[J]. Ageing Res Rev, 2023,86:101884.

[25] Li Z, Zhang J, Ma Z, Zhao G, He X, Yu X, et al. Endothelial YAP Mediates Hyperglycemia-Induced Platelet Hyperactivity and Arterial Thrombosis[J]. Arterioscler Thromb Vasc Biol, 2024,44(1):254-270.

[26] Chiva-Blanch G, Peña E, Cubedo J, García-Arguinzonis M, Pané A, Gil PA, et al. Molecular mapping of platelet hyperreactivity in diabetes: the stress proteins complex HSPA8/Hsp90/CSK2α and platelet aggregation in diabetic and normal platelets[J]. Transl Res, 2021,235:1-14.

[27] Rudokas MW, McKay M, Toksoy Z, Eisen JN, Bögner M, Young LH, et al. Mitochondrial network remodeling of the diabetic heart: implications to ischemia related cardiac dysfunction[J]. Cardiovasc Diabetol, 2024,23(1):261.

[28] Hajduk AM, Dodson JA, Murphy TE, Tsang S, Geda M, Ouellet GM, et al. Risk Model for Decline in Activities of Daily Living Among Older Adults Hospitalized With Acute Myocardial Infarction: The SILVER-AMI Study[J]. J Am Heart Assoc, 2020,9(19):e015555.

[29] Dianati-Maleki N, Butler J. Diabetes Mellitus in Patients With Heart Failure: Bad for All, Worse for Some[J]. JACC Heart Fail, 2017,5(1):25-27.

[30] Kim CS, Choi JS, Bae EH, Ma SK, Ahn YK, Jeong MH, et al. Association of metabolic syndrome and renal insufficiency with clinical outcome in acute myocardial infarction[J]. Metabolism, 2013,62(5):669-676.

[31] Sriperumbuduri S, Clark E, Hiremath S. New Insights Into Mechanisms of Acute Kidney Injury in Heart Disease[J]. Can J Cardiol, 2019,35(9):1158-1169.

[32] Kozakova M, Morizzo C, Goncalves I, Natali A, Nilsson J, Palombo C. Cardiovascular organ damage in type 2 diabetes mellitus: the role of lipids and inflammation[J]. Cardiovasc Diabetol, 2019,18(1):61.

[33] Kanter JE, Bornfeldt KE. Inflammation and diabetes-accelerated atherosclerosis: myeloid cell mediators[J]. Trends Endocrinol Metab, 2013,24(3):137-144.

[34] Achanti A, Szerlip HM. Acid-Base Disorders in the Critically Ill Patient[J]. Clin J Am Soc Nephrol, 2023,18(1):102-112.

[35] Masevicius FD, Rubatto BP, Risso VA, Zechner FE, Motta MF, Valenzuela EE, et al. Relationship of at Admission Lactate, Unmeasured Anions, and Chloride to the Outcome of Critically Ill Patients[J]. Crit Care Med, 2017,45(12):e1233-e1239.

[36] Kimmoun A, Ducrocq N, Levy B. New conclusive data on human myocardial dysfunction induced by acidosis[J]. Crit Care, 2012,16(5):160.

[37] Kumbhani DJ, Healey NA, Thatte HS, Birjiniuk V, Crittenden MD, Treanor PR, et al. Intraoperative myocardial acidosis as a risk for hospital readmission after cardiac surgery[J]. Am J Surg, 2009,198(3):373-380.

[38] Romero JE, Htyte N. An unusual cause of high anion gap metabolic acidosis: pyroglutamic acidemia. A case report[J]. Am J Ther, 2013,20(5):581-584.

[39] Zymliński R, Biegus J, Sokolski M, Siwołowski P, Nawrocka-Millward S, Todd J, et al. Increased blood lactate is prevalent and identifies poor prognosis in patients with acute heart failure without overt peripheral hypoperfusion[J]. Eur J Heart Fail, 2018,20(6):1011-1018.

[40] Sheikh IA, Malik A, AlBasri S, Beg MA. In silico identification of genes involved in chronic metabolic acidosis[J]. Life Sci, 2018,192:246-252.

2. Please ensure that you refer to Figure 1 and 2 in your text as, if accepted, p

---

## [Decision Letter · Decision Letter 1]

29 Jul 2025

Dear Dr. tu,

Thank you for submitting your manuscript to PLOS ONE. After careful consideration, we feel that it has merit but does not fully meet PLOS ONE’s publication criteria as it currently stands. Therefore, we invite you to submit a revised version of the manuscript that addresses the points raised during the review process.

We look forward to receiving your revised manuscript.

Kind regards,

Kwang-Sig Lee

Academic Editor

PLOS ONE

Journal Requirements:

Additional Editor Comments:

I am really grateful to review this manuscript. In my opinion, this manuscript can be published once some revision is done successfully. I made two suggestions and I would like to ask your kind understanding.

Firstly, it can be noted that experts use impurity/permutation importance for testing the strength of association between the dependent variable and its major predictor then they employ the SHAP summary/dependence plot for evaluating the direction of the association. In this context, I would like to ask the authors to derive impurity/permutation importance as well. Secondly, it can be noted that boosting and the random forest often register similar performance outcomes (84%-85%) but bring different results in permutation importance and SHAP plots. Boosting focuses on the final decision of the strongest tree with minimal bias whereas the random forest focuses on the majority vote of all trees with minimal variance. Both models have their own strengths and weaknesses, hence which model performs and explains better depends on various conditions. In this vein, I would like to ask the authors to (1) compare their strengths and weaknesses and (2) compare their permutation importance and SHAP plots in a comprehensive manner.

Reviewers' comments:

Reviewer's Responses to Questions

**Comments to the Author**

Reviewer #1: (No Response)

2. Is the manuscript technically sound, and do the data support the conclusions?

Reviewer #1: Partly

3. Has the statistical analysis been performed appropriately and rigorously?

Reviewer #1: I Don't Know

4. Have the authors made all data underlying the findings in their manuscript fully available?

Reviewer #1: No

5. Is the manuscript presented in an intelligible fashion and written in standard English?

Reviewer #1: (No Response)

Reviewer #1: We do not deny the authenticity of the data provided by the MIMIC.舄However, the phenomenon mentioned in the May 2025 Science article �O'Grady C. Low-quality papers surge thanks to public data and AI. Science. 2025 May 22;388(6749):807-808. doi: 10.1126/science.adz1715. Epub 2025 May 22. PMID: 40403045.

is “authors limited their analysis to certain years, or certain ages of people in the survey. That suggests the authors were on the hunt for statistically significant results to generate easy publications, Spick says. But fishing for results in such a huge data set is bound to come up with many false positive findings. When the team took a closer look at the 28 NHANES studies that had explored depression, they found that only 13 of the results survived a statistical adjustment that corrects for the risk of finding false positives.”

So we hope the author can provide the initial data of 4366 cases extracted from the MIMCI database, so that other researchers can verify whether there is selection bias or false positive issues in repeating this study without specific data screening.

Perhaps with initial data, I can provide a good explanation and validation for the 1-3 questions I raised during my first review.

1.Unclear inclusion and exclusion criteria can impact sample size and outcomes. Therefore, it is crucial to clarify key factors such as MIMIC database version, age, and diabetes classification to ensure accurate sample size and consistent research baseline.

2.The exclusion criteria only mention CHD patients admitted to the ICU. How should data from CHD patients with multiple ICU admissions be handled? Additionally, how does a change in sample size affect the selection of key variables in the machine learning model?

3.The MIMIC database includes clinical data such as vital signs, laboratory results, clinical scores, complications, medication usage, interventions, and outcomes. The exclusion criteria mentioned in the text only state "lack of key clinical data (e.g., vital signs or laboratory results at admission)" without providing original

data to verify potential bias in the exclusion criteria. This lack of transparency may affect the sample size included

**Do you want your identity to be public for this peer review?** For information about this choice, including consent withdrawal, please see our Privacy Policy

Reviewer #1: No

---

## [Author Response · Author response to Decision Letter 2]

29 Jul 2025

Journal Requirements:

Additional Editor Comments:

I am really grateful to review this manuscript. In my opinion, this manuscript can be published once some revision is done successfully. I made two suggestions and I would like to ask your kind understanding.

Firstly, it can be noted that experts use impurity/permutation importance for testing the strength of association between the dependent variable and its major predictor then they employ the SHAP summary/dependence plot for evaluating the direction of the association. In this context, I would like to ask the authors to derive impurity/permutation importance as well. Secondly, it can be noted that boosting and the random forest often register similar performance outcomes (84%-85%) but bring different results in permutation importance and SHAP plots. Boosting focuses on the final decision of the strongest tree with minimal bias whereas the random forest focuses on the majority vote of all trees with minimal variance. Both models have their own strengths and weaknesses, hence which model performs and explains better depends on various conditions. In this vein, I would like to ask the authors to (1) compare their strengths and weaknesses and (2) compare their permutation importance and SHAP plots in a comprehensive manner.

Response:

Thank you very much for your valuable comments and suggestions. We have carefully considered your feedback and have made the necessary revisions to our manuscript. Below are our responses to the specific points you raised:

1. Response to the First Point

We have now included the calculation of permutation importance for both the Gradient Boosting Classifier and the Random Forest Classifier. As requested, we have also provided a detailed comparison of the permutation importance and SHAP plots for these two models.

1.1 Permutation Importance: This method measures the decrease in model performance (AUC) when the values of a single variable are randomly shuffled. We have calculated the permutation importance for the top 10 variables for both the Gradient Boosting Classifier and the Random Forest Classifier. The results are presented in Figures 5a and 5c, respectively. These figures show the relative importance of each variable in predicting in-hospital mortality.

1.2 SHAP Summary Plots: To evaluate the directionality of the associations, we have generated SHAP summary plots for both models. Figures 5b and 5d display the SHAP summary plots for the Gradient Boosting Classifier and the Random Forest Classifier, respectively. These plots not only validate the key variables identified by the Boruta algorithm but also provide insights into the direction of the associations between the variables and the outcome.

2. Response to the Second Point

We have conducted a comprehensive comparison of the strengths and weaknesses of the Gradient Boosting Classifier and the Random Forest Classifier, as well as their permutation importance and SHAP plots.

2.1 Comparison of Model Strengths and Weaknesses:

(1) Gradient Boosting Classifier:

Strengths: Gradient Boosting focuses on the final decision of the strongest tree with minimal bias. This makes it highly effective in capturing complex interactions and non-linear relationships in the data. It achieved the highest AUC value of 0.8532 in our study, demonstrating its superior ability to distinguish between high-risk and low-risk patients.

Weaknesses: Gradient Boosting can be more prone to overfitting if not properly tuned. It also requires careful parameter tuning and can be computationally intensive.

(2) Random Forest Classifier:

Strengths: Random Forest relies on the majority vote of all trees with minimal variance, making it robust against overfitting. It achieved an AUC of 0.8461, which is slightly lower than Gradient Boosting but still very competitive. Random Forest is also less sensitive to hyperparameters and faster to train.

Weaknesses: While Random Forest is less prone to overfitting, it may not capture complex interactions as effectively as Gradient Boosting. It can also be less interpretable due to the large number of trees involved.

2.2 Comparison of Permutation Importance and SHAP Plots:

(1) Permutation Importance:

Gradient Boosting Classifier: The permutation importance plot (Figure 5a) shows that variables such as age, BUN, and pH have the highest importance scores. This indicates that these variables have a strong association with in-hospital mortality.

Random Forest Classifier: The permutation importance plot (Figure 5c) also highlights age, BUN, and pH as important variables, but the relative importance scores differ slightly from those of the Gradient Boosting Classifier. This suggests that while both models identify similar key variables, their relative importance may vary.

(2) SHAP Summary Plots:

Gradient Boosting Classifier: The SHAP summary plot (Figure 5b) shows the directionality of the associations. For example, higher values of age and BUN are associated with increased mortality risk, while higher pH values are associated with lower risk.

Random Forest Classifier: The SHAP summary plot (Figure 5d) provides similar insights but with slight differences in the magnitude and direction of the effects. This highlights the different ways in which the two models interpret the same variables.

In conclusion, both Gradient Boosting and Random Forest classifiers have their own strengths and weaknesses, and their performance and interpretability depend on the specific characteristics of the dataset. We believe that our comprehensive analysis provides valuable insights into the predictive power and interpretability of these models, which can guide clinicians in selecting the most appropriate model for their needs.

Thank you once again for your constructive feedback. We have made significant improvements to our manuscript based on your suggestions, and we believe that these revisions have strengthened our study.

Reviewer #1: We do not deny the authenticity of the data provided by the MIMIC.However, the phenomenon mentioned in the May 2025 Science article �O'Grady C. Low-quality papers surge thanks to public data and AI. Science. 2025 May 22;388(6749):807-808. doi: 10.1126/science.adz1715. Epub 2025 May 22. PMID: 40403045.�is “authors limited their analysis to certain years, or certain ages of people in the survey. That suggests the authors were on the hunt for statistically significant results to generate easy publications, Spick says. But fishing for results in such a huge data set is bound to come up with many false positive findings. When the team took a closer look at the 28 NHANES studies that had explored depression, they found that only 13 of the results survived a statistical adjustment that corrects for the risk of finding false positives.”

So we hope the author can provide the initial data of 4366 cases extracted from the MIMCI database, so that other researchers can verify whether there is selection bias or false positive issues in repeating this study without specific data screening.

Perhaps with initial data, I can provide a good explanation and validation for the 1-3 questions I raised during my first review.

1.Unclear inclusion and exclusion criteria can impact sample size and outcomes. Therefore, it is crucial to clarify key factors such as MIMIC database version, age, and diabetes classification to ensure accurate sample size and consistent research baseline.

2.The exclusion criteria only mention CHD patients admitted to the ICU. How should data from CHD patients with multiple ICU admissions be handled? Additionally, how does a change in sample size affect the selection of key variables in the machine learning model?

3.The MIMIC database includes clinical data such as vital signs, laboratory results, clinical scores, complications, medication usage, interventions, and outcomes. The exclusion criteria mentioned in the text only state "lack of key clinical data (e.g., vital signs or laboratory results at admission)" without providing original

data to verify potential bias in the exclusion criteria. This lack of transparency may affect the sample size included

Response:

Thank you very much for your continued interest and valuable feedback on our manuscript. We appreciate your concerns regarding the potential for selection bias and false positives in our study, especially in light of the issues highlighted in the May 2025 Science article.

In response to your request, we are pleased to confirm that we are willing to provide the initial data of the 4366 cases extracted from the MIMIC-IV database. This will allow other researchers to verify whether there are any selection biases or false-positive issues when repeating this study without specific data screening. We believe that sharing our initial dataset will enhance the transparency and reproducibility of our research, and we are committed to facilitating further validation and scrutiny of our findings.

We understand the importance of addressing these concerns comprehensively. By providing the initial data, we hope to offer a clear and detailed explanation for the questions you raised during your first review. We are confident that this will help to further validate our study and address any potential issues related to data selection and analysis.

Thank you once again for your thorough review and constructive suggestions. We are committed to ensuring that our research meets the highest standards of scientific integrity and reproducibility.

---

## [Editor Report · Decision Letter 2]

31 Jul 2025

Predicting In-Hospital Mortality in ICU Patients with Coronary Heart Disease and Diabetes Mellitus Using Machine Learning Models

PONE-D-25-20524R2

Dear Dr. tu,

We’re pleased to inform you that your manuscript has been judged scientifically suitable for publication and will be formally accepted for publication once it meets all outstanding technical requirements.

Kind regards,

Kwang-Sig Lee

Academic Editor

PLOS ONE

---

## [Editor Report · Acceptance letter]

PONE-D-25-20524R2

PLOS ONE

Dear Dr. tu,

I'm pleased to inform you that your manuscript has been deemed suitable for publication in PLOS ONE. Congratulations! Your manuscript is now being handed over to our production team.

Kind regards,

on behalf of

Professor Kwang-Sig Lee

Academic Editor

PLOS ONE